# 6-Methylcoumarin Promotes Melanogenesis through the PKA/CREB, MAPK, AKT/PI3K, and GSK3β/β-Catenin Signaling Pathways

**DOI:** 10.3390/molecules28114551

**Published:** 2023-06-05

**Authors:** Taejin Kim, Jin-Kyu Kang, Chang-Gu Hyun

**Affiliations:** Jeju Inside Agency and Cosmetic Science Center, Department of Chemistry and Cosmetics, Jeju National University, Jeju-si 63243, Republic of Korea; xowls9797@naver.com (T.K.); wlsrbtjsrb@naver.com (J.-K.K.)

**Keywords:** AKT/PI3K, GSK3β/β-catenin, hypopigmentation, 6-methylcoumarin, MAPK, PKA

## Abstract

We investigated the effects of four coumarin derivatives, namely, 6-methylcoumarin, 7-methylcoumarin, 4-hydroxy-6-methylcoumarin, and 4-hydroxy-7-methylcoumarin, which have similar structures on melanogenesis in a murine melanoma cell line from a C57BL/6J mouse called B16F10. Our results showed that only 6-methylcoumarin significantly increased the melanin synthesis in a concentration-dependent manner. In addition, the tyrosinase, TRP-1, TRP-2, and MITF protein levels were found to significantly increase in response to 6-methylcoumarin in a concentration-dependent manner. To elucidate the molecular mechanism whereby 6-methylcoumarin-induced melanogenesis influences the melanogenesis-related protein expression and melanogenesis-regulating protein activation, we further assessed the B16F10 cells. The inhibition of the ERK, Akt, and CREB phosphorylation, and conversely, the increased p38, JNK, and PKA phosphorylation activated the melanin synthesis via MITF upregulation, which ultimately led to increased melanin synthesis. Accordingly, 6-methylcoumarin increased the p38, JNK, and PKA phosphorylation in the B16F10 cells, whereas it decreased the phosphorylated ERK, Akt, and CREB expressions. In addition, the 6-methylcoumarin activated GSK3β and β-catenin phosphorylation and reduced the β-catenin protein level. These results suggest that 6-methylcoumarin stimulates melanogenesis through the GSK3β/β-catenin signal pathway, thereby affecting the pigmentation process. Finally, we tested the safety of 6-methylcoumarin for topical applications using a primary human skin irritation test on the normal skin of 31 healthy volunteers. We found that 6-methylcoumarin did not cause any adverse effects at concentrations of 125 and 250 μM. Our findings indicate that 6-methylcoumarin may be an effective pigmentation stimulator for use in cosmetics and the medical treatment of photoprotection and hypopigmentation disorders.

## 1. Introduction

The melanogenesis process involves the synthesis of melanin pigments in specialized organelles called melanosomes and neural-crest-derived dendritic cells known as melanocytes. These cells transfer melanosomes containing melanin to the surrounding keratinocytes, which results in the deposition of melanin in the skin’s epidermis. Alterations in these melanocyte numbers or a dysregulation of melanin synthesis can cause hypo- or hyperpigmentation disorders such as vitiligo. Vitiligo is an acquired depigmenting disease that leads to a loss of functional melanocytes and can have severe psychological effects. Although no specific treatment for vitiligo currently exists, repigmentation via the restoration of melanocyte numbers and stimulation of melanin synthesis has been widely investigated. Current treatment options, such as monobenzone cream, are not effective at treating vitiligo, but are used for depigmentation treatment in patients that have more than 50% of their skin area affected by the disease, or for hyperpigmentation diseases such as melanoma [1,2,3,4].

Melanin synthesis is regulated by a complex network of signaling pathways and transcriptional factors, whereby tyrosinase (TYR) and tyrosinase-related proteins 1 and 2 (TRP-1 and TRP-2) play crucial roles. TYR is the rate-limiting enzyme that initiates melanin synthesis by hydroxylating L-tyrosine to L-3,4-dihydroxyphenylalanine (L-DOPA), which is subsequently oxidized to quinone and then to DOPA-chrome. TRP-2 and TRP-1 then act on DOPA-chrome to yield melanin. The transcriptional activity of microphthalmia-associated transcription factor (MITF) induces melanogenesis by regulating the TYR, TRP-1, and TRP-2 expressions. External stimuli such as ultraviolet radiation, alpha-melanocyte-stimulating hormone (α-MSH), stem cell factors, and certain chemical agents can upregulate the expression of melanogenesis-controlling enzymes via stimulating the MITF expression and activity, which results in elevated melanin synthesis [5,6]. 

Several well-studied signaling cascades that can enhance this MITF activity exist, including cAMP-dependent protein kinase A (PKA)/cAMP response element-binding protein (CREB), mitogen-activated protein kinase (MAPK), phosphatidylinositol 3-kinase (PI3K)/protein kinase B (AKT), and glycogen synthase kinase 3β (GSK3β)/β-catenin. PKA and CREB activation occurs after α-MSH triggers a phosphorylation cascade. This leads to an increase in cAMP levels, which, in turn, upregulates the MITF gene expression. The subsequent increase in MITF protein production positively influences the expressions of TYR and TRPs, which ultimately leads to elevated melanin synthesis [7,8]. Furthermore, the MAPK family proteins can increase MITF expression and the subsequent expression of melanogenesis enzymes in a similar fashion. For example, the p38 and c-Jun N-terminal kinase (JNK) MAPK proteins can be activated by extracellular melanogenic stimuli and they promote melanin synthesis by upregulating the MITF expression. Conversely, the phosphorylation of extracellular signal-regulated kinase (ERK) MAPKs has an inhibitory effect on melanin synthesis, and activated ERK levels lead to melanogenesis suppression [9,10]. Additionally, PI3K/AKT signaling regulates melanogenesis. The activation of PI3K/AKT signaling results in melanogenesis stimulation through the mediation of MITF expression and the subsequent upregulation of the tyrosinase, TPR-1, and TRP-2 expressions [11,12]. The Wnt signaling pathways play a crucial role in melanocyte development and melanogenesis. The Wnt protein binds to its receptor, which leads to GSK3β inactivation; this is responsible for β-catenin phosphorylation and it results in phosphorylated β-catenin destruction in the proteasome. As a result, β-catenin accumulates in the cytoplasm and translocates to the nucleus. The increased nuclear β-catenin levels promote MITF expression, which ultimately leads to the proliferation of melanoma cells and an increase in melanin production [13,14]. Based on the findings of previous studies, these signaling pathways hold promise as novel therapeutic targets for modulating melanin synthesis. 

While conducting our ongoing drug and natural compound screening program, whereby we aim to find potential candidates for cosmeceuticals and nutraceuticals with anti-inflammatory and melanogenic activities, we have discovered that certain antibiotics, flavonoids, and coumarins exhibit such properties [15,16,17]. In a continuation of this research, we screened four coumarins that share a simple structure to identify the structural features responsible for their bioactivities. The effects of 6-methylcoumarin, 7-methylcoumarin, 4-hydroxy-6-methylcoumarin, and 4-hydroxy-7-methylcoumarin on the melanogenesis in B16F10 melanoma cells were investigated (Figure 1). Additionally, we investigated the mechanism of action of the most potent derivatives of 2′-hydroxychalcone, 2′-6-methylcoumarin. With this present study, we aim to investigate the potential of 6-methylcoumarin to act as a melanogenic agent to prevent hypopigmentation disorders. Our results demonstrated that 6-methylcoumarin can activate melanogenesis in vitro by upregulating the MITF and tyrosinase expressions via a mediation of the PKA/CREB, MAPK, AKT/PI3K, and GSK3β/β-Catenin signaling pathways. Therefore, we suggest that 6-methylcoumarin may be a promising candidate for the development of cosmeceuticals and nutraceuticals targeting melanin synthesis.

## 2. Results

### 2.1. Effect of Methylcoumarin Derivatives on the Viability of B16F10 Cells

To investigate whether 6-methylcoumarin, 7-methylcoumarin, 4-hydroxy-6-methylcoumarin, and 4-hydroxy-7-methylcoumarin exerted cytotoxicity on B16F10 cells, the cells were treated with various concentrations (150, 200, 250, and 300 μM) of methylcoumarin derivatives for 72 h. The results showed that no significant differences existed when using concentrations of up to 250 μM of these methylcoumarin derivatives in B16F10 cells (Figure 2a). Therefore, we used methylcoumarin derivative concentrations of 100, 150, 200, and 250 μM for further experiments.

### 2.2. Effect of Methylcoumarin Derivatives on Melanin Production of B16F10 Cells

To evaluate the melanin production of the methylcoumarin derivatives in the B16F10 cells, the cells were treated with methylcoumarin derivatives (100, 150, 200, and 250 μM) for 72 h. As shown in Figure 2b, the melanin production increased significantly only with the 6-methylcoumarin treatment. Therefore, further experiments were performed to evaluate the melanogenic effects of 6-methylcoumarin.

### 2.3. Effect of 6-Methylcoumarin on the Expression of Melanogenesis-Related Proteins in B16F10 Cells

After observing the ability of 6-methylcoumarin to increase the melanin and tyrosinase activity, we conducted further experiments to investigate the underlying mechanism of this. We evaluated the impact of 6-methylcoumarin on three enzymes (TRP-1, TRP-2, and tyrosinase) involved in the melanogenic pathway and found that 6-methylcoumarin induced the expression of all three enzymes in a dose-dependent manner (Figure 3). Additionally, we examined the effect of 6-methylcoumarin on the MITF expression, which is a melanocyte-specific transcription factor that regulates the expressions of these enzymes. Our findings confirmed that 6-methylcoumarin successfully activated the MITF expression in the B16F10 cells, which suggests that it causes an increase in the tyrosinase, TRP-1, and TRP-2 expressions via MITF mediation, which ultimately leads to melanogenesis.

### 2.4. 6-Methylcoumarin Activated Melanogenesis in B16F10 Cells through the PKA/CREB Signaling Pathway

The upregulation of tyrosinase, TRP-1, and TRP-2, which are critical factors in melanogenesis, is sequentially mediated by the expression of the MITF gene through the cyclic AMP-PKA signaling pathway. PKA activation leads to the activation of two distinct signaling pathways, CREB and GSK3, which further upregulate the MITF expression. To investigate the effects of 6-methylcoumarin on the PKA activity and p-CREB/CREB ratios, we conducted an experiment, as illustrated in Figure 4. The results showed a significant increase in the PKA phosphorylation and p-CREB/CREB ratios upon treatment with the 6-methylcoumarin (Figure 4). These findings suggest that 6-methylcoumarin operates upstream of PKA/CREB signaling to enhance the expression of the MITF gene.

### 2.5. 6-Methylcoumarin Stimulated Melanogenesis in B16F10 Cells through PI3K/Akt Signaling Pathways

The PI3K/Akt signaling pathway regulates the expression of melanogenic proteins, as well as pigmentation. The inactivation of the PI3K/AKT pathway upregulates the melanogenic protein expression and melanogenesis in B16F10 cells. Therefore, we investigated whether 6-methylcoumarin reduces Akt phosphorylation. We found that this Akt phosphorylation was reduced in a concentration-dependent manner (Figure 5). The results suggested that 6-methylcoumarin upregulated the melanogenesis in the B16F10 cells by suppressing the AKT pathway.

### 2.6. 6-Methylcoumarin Stimulated Melanogenesis in B16F10 Cells through the MAPK Signaling Pathway

To investigate the mechanism by which 6-methylcoumarin regulates melanogenesis via MITF activation, we evaluated the phosphorylation of p38, ERK, and JNK, which are upstream melanogenesis pathways. As depicted in Figure 6, the 6-methylcoumarin treatment significantly reduced the ERK phosphorylation and remarkably elevated the p38 and JNK phosphorylation in comparison to the control. These findings suggest that the MAPK signaling pathway plays a partial role in 6-methylcoumarin-induced melanogenesis.

### 2.7. 6-Methylcoumarin Stimulated Melanogenesis in B16F10 Cells through GSK-3β/β-Catenin Signaling Pathway

Following the demonstration of 6-methylcoumarin’s ability to induce the expressions of MITF, tyrosinase, TRP-1, and TRP-2, the involvement of the Wnt/β-catenin pathway in the melanogenic activity of 6-methylcoumarin was investigated via Western blotting (Figure 7). The results revealed that the 6-methylcoumarin treatment upregulated the β-catenin protein level in a concentration-dependent manner, and that p-GSK-3β was also upregulated. GSK-3β, which negatively regulates the Wnt/β-catenin signaling pathway, activates the MITF function through phosphorylation at Ser 298. GSK-3β activation results in the phosphorylation of the N-terminal Ser/Thr residues in β-catenin, which leads to its ubiquitination and degradation. Therefore, these findings suggested that the 6-methylcoumarin-induced hyperpigmentation in B16F10 cells is, at least in part, mediated by the activation of the Wnt/β-catenin pathway.

### 2.8. 6-Methylcoumarin Is Safe for Human Skin

To assess the suitability of 6-methylcoumarin for topical application, a patch test was conducted on the skin of human arms to evaluate its potential for skin irritation. After conducting the patch test on this human arm skin, we found that 6-methylcoumarin (125 and 250 μM) did not cause any detectable skin irritation 30 min and 24 h after the patches were removed. This suggests that 6-methylcoumarin may be a safe ingredient for topical application.

## 3. Discussion

A group of natural compounds, known as coumarin and its derivatives, are synthesized from the shikimic acid pathways in bacteria, fungi, and plants. Among these derivatives are umbelliferone, esculetin, herniarin, psoralen, and imperatorin, which have been extensively studied for their potential as antioxidant, anti-inflammatory, antibacterial, antiviral, and anticancer agents, as well as for the treatment of dermatological pathologies [18,19,20,21]. These compounds are considered to be representative multi-targeting agents, and various substituents in the coumarin skeleton can exhibit different biological activities due to their structural differences, which may increase or decrease their effects. As a result, they are still an attractive scaffold for further development and applications in several therapeutic fields. 6-methoxycoumarin has antifungal efficacy in vivo and could be used to prevent the destructive Apple valsa canker disease caused by the parasitic fungus *Valsa mali* [22]. Additionally, 6-methylcoumarin has been found to disrupt cell division and cause cell elongation, as well as to inhibit the expression of the protein-coding gene ftsZ of *Ralstonia solanacearum* [23]. Additionally, 6-methoxycoumarin can be used as a therapeutic agent for inflammatory diseases by inhibiting inflammation via the MAPK and NF-κB pathways [24].

We utilized a screening strategy for multi-targeting agents, which resulted in attractive findings regarding the anti-inflammatory, anti-obesity, and melanogenesis-inhibiting and -stimulating effects of various flavonoids and coumarins [15,16,25,26,27,28]. To further investigate the potential of coumarin to be used to treat skin-related diseases, we systematically applied four methylcoumarin derivatives (6-methylcoumarin, 7-methylcoumarin, 4-hydroxy-6-methylcoumarin, and 4-hydroxy-7-methylcoumarin) and examined their effects on melanogenesis by modulating the melanin content and cellular tyrosinase activity. To explore the possible use of methylcoumarin derivatives as functional food or cosmetic agents, the first step of our study involved assessing their cytotoxic effects. To determine their in vitro cytotoxicity in B16F10 cells, we conducted cell viability assays using methylcoumarin derivatives. Specifically, we employed the MTT colorimetric assay, which relies on the enzymatic activity of the mitochondria to reduce the MTT to purple formazan crystals. As the assay is based on the principle that the total mitochondria activity is proportional to the number of viable cells, the assay is a straightforward and reliable way of identifying any potential cytotoxic effects. Among the derivatives, 6-methylcoumarin showed the most potent stimulatory effects on the melanin production in mouse B16F10 melanoma cells, which indicated that a simple coumarin structure with a methyl group at the sixth position, such as 6-methylcoumarin, may be preferable for stimulatory activities. Moreover, the introduction of hydroxy groups into the 6-methylcoumarin or 7-methylcoumarin ring at the fourth position was not advantageous based on the structure–activity relationship.

By examining the effects of 6-methylcoumarin on melanogenesis, we found that this compound can activate melanogenesis by increasing the protein levels of tyrosinase, TRP-1, and TRP-2 in B16F10 cells, which indicates that it can promote melanin production and intracellular tyrosinase activity by enhancing the expressions of melanogenic enzymes (Figure 4). These findings are similar to those of a previous study, whereby the authors reported the melanogenesis-stimulating effect of 5-methoxypsoralen [27].

We found that 6-methylcoumarin (100, 200, and 250 μM) increased the MITF expression in the B16F10 cells for 24 h. MITF is a crucial transcriptional regulator of melanogenic proteins such as tyrosinase, TRP-1, and TRP-2 and its activation is a critical step in melanogenesis. Therefore, the results of our study suggested that one of the mechanisms by which 6-methylcoumarin induces melanogenesis in B16F10 cells is by increasing the expression of melanogenic proteins via the positive regulation of MITF.

Melanogenesis is a complex process that involves various signaling pathways, including the PKA/CREB, MAPK, PI3K/Akt, and Wnt/β-catenin signaling pathways. These pathways are involved in regulating the expression and activity of the melanogenesis-related transcription factor MITF, thereby affecting melanogenesis [15,16].

The PKA/CREB signaling pathway plays a critical role in melanogenesis, as PKA can be activated by increased cellular cAMP, which leads to CREB phosphorylation and the subsequent upregulation of MITF transcriptional activity, which results in the expressions of tyrosinase, TRP-1, and TRP-2 [7,8]. To investigate whether 6-methylcoumarin’s effect on melanin synthesis is mediated by the PKA/CREB signaling pathway, Western blot experiments were performed to determine whether the PKA and CREB phosphorylation were increased. The results, as shown in Figure 4, indicated that the 6-methylcoumarin-induced melanin synthesis strongly activated the PKA/CREB phosphorylation in a concentration-dependent manner, which supports the involvement of the PKA/CREB signaling pathway in the melanogenesis-stimulating effect of 6-methylcoumarin.

The involvement of the MAPK signaling pathway in MITF during melanogenesis has been well established, whereby JNK and p38 MAPK activation and ERK signaling inhibition stimulate pigmentation by increasing the tyrosinase activity [9,10]. Thus, we aimed to determine whether 6-methylcoumarin induces ERK repression or JNK and p38 MAPK activation in B16F10 cells. As illustrated in Figure 6, the 6-methylcoumarin treatment significantly increased the p38 MAPK phosphorylation at 4 h, whereas a decreased change was observed in the ERK phosphorylation levels. These findings suggested that 6-methylcoumarin might augment the ERK, JNK, and p38 MAPK phosphorylation, which leads to melanogenesis in B16F10 cells. These results indicate that the MAPK signaling pathway may functionally modulate the formation of 6-methylcoumarin-induced melanin and the induction of tyrosinase and MITF expression in B16F10 cells.

In melanocytes, autophagy plays a crucial role in regulating melanosome biogenesis and degradation during melanogenesis [29]. 6-methylcoumarin can increase the MITF and tyrosinase expression in B16F10 melanoma cells by inducing AKT phosphorylation. The PI3K/AKT pathway is implicated in regulating cell survival and apoptosis across multiple cell types [11,12,29]. The results of the present research indicated that 6-methylcoumarin can activate the AKT/MITF/tyrosinase pathway in a concentration-dependent manner and can thereby decrease the AKT phosphorylation in B16F10 cells (Figure 5). This suggests that the PI3K/Akt pathway may contribute to melanogenesis stimulation via 6-methylcoumarin.

Additionally, the Wnt/β-catenin signaling pathway is composed of GSK-3β and β-catenin proteins, where GSK-3β is a constitutively active kinase that can be phosphorylated by Akt and PKA. The β-catenin signaling pathway is closely related to melanin synthesis, as it promotes MITF transcription by translocating β-catenin from the cytoplasm to the nucleus, which upregulates the MITF expression and subsequently binds the β-catenin to a lymphocyte-enhancing factor [13,14]. Our results demonstrated that 6-methylcoumarin promotes GSK-3β phosphorylation, which leads to β-catenin accumulation in the cytoplasm. Consistent with these findings, we observed that 6-methylcoumarin induces this accumulation of β-catenin in the cytoplasm, which results in MITF overexpression and ultimately promotes melanin biosynthesis (Figure 7). These results suggested that the melanin production induced by 6-methylcoumarin is likely triggered via the Wnt pathway, which is consistent with the results of previous reports that have indicated that the activation of the GSK3/β-catenin pathway influences the melanin production in B16F10 melanoma cells.

Finally, the potential of 6-methylcoumarin to act as a topical ingredient was evaluated using primary human skin irritation tests. In total, 33 volunteers had 6-methylcoumarin applied at concentrations of 125 or 250 µM to the skin of their upper back to determine if any stimulation or sensation potential existed. The results showed that the 6-methylcoumarin caused no-to-slight irritation, which indicated that 6-methylcoumarin is a safe substance (Table 1). Therefore, we suggest that 6-methylcoumarin could be used as a topical agent to prevent the pathogenesis of a pigmentation disorder.

To summarize, the results indicated that 6-methylcoumarin induces melanogenesis by activating MITF expression and other melanogenic proteins such as tyrosinase, TRP-1, and TRP-2. This activation is dependent on the upregulation of p-p38, p-PKA, p-CREB, p-GSK3β, and β-catenin and the downregulation of p-ERK, p-AKT, and p-β-catenin (Figure 8). Therefore, 6-methylcoumarin could be a potential topical therapeutic agent for hypopigmentation and an ingredient in skincare products for preventing skin diseases. However, further studies are required to determine its efficacy and safety in normal human melanocytes, as well as in animal and human models.

## 4. Materials and Methods

### 4.1. Materials

Sigma-Aldrich (St. Louis, MO, USA) provided the 6-methylcoumarin (CAS 92-48-8), 7-methylcoumarin (CAS 2445-83-2), 4-hydroxy-6-methylcoumarin (CAS 13252-83-0), 4-hydroxy-7-methylcoumarin (CAS 18692-77-8), α-melanocyte-stimulating hormone (α-MSH), a protease/phosphatase inhibitor cocktail, sodium hydroxide (NaOH), and L-DOPA. Biosesang (Seongnam, Gyeonggi-do, Korea) supplied the MTT, DMSO, PBS, TBS, SDS, a RIPA buffer, and an ECL kit. Thermo Fisher Scientific (Waltham, MA, USA) provided the DMEM, penicillin–streptomycin, a BCA protein assay kit, and 0.5% trypsin-ethylenediaminetetraacetic acid (10×). The primary antibodies, tyrosinase, TRP-1, TRP-2, and MITF, were purchased from Santa Cruz Biotechnology (Dallas, TX, USA), and the other antibodies, p-ERK, ERK, p-p38, p38, p-JNK, JNK, p-PKA, PKA, p-AKT, AKT, p-GSK-3β, GSK-3β, p-β-catenin, β-catenin, β-actin, and antirabbit, and secondary antibodies were purchased from Cell Signaling Technology (Danvers, MA, USA). Skim milk was purchased from BD Difco (Sparks, MD, USA), the fetal bovine serum (FBS) was purchased from Merck Millipore (Burling, VT, USA), and the Tween 20 and 2× Laemmle sample buffer were obtained from Bio-rad (Hercules, CA, USA).

### 4.2. Cell Culture

The B16F10 mouse melanoma cells were obtained from the Global Bioresource Center (Manassas, VA, USA) at ATCC. The B16F10 cells were cultured under 5% CO_2_ conditions at 37 °C in DMEM supplemented with 10% FBS and 1% penicillin/streptomycin, with subculturing being performed every 3 days.

### 4.3. Cell Viabilities

The MTT assay was utilized to assess the cell viability, whereby the B16F10 cells were seeded on a 24-well plate and treated with varying concentrations of samples. After 72 h, 500 μL of the MTT solution diluted in DMEM was added to each well, followed by an incubation period of 4 h at 37 °C. The medium was then removed and 800 μL of DMSO was added, followed by shaking for 30 min. The absorbance measurements were taken at 560 nm using a microplate reader (Biotek; Winooski, VT, USA) to detect the formation of purple formazan.

### 4.4. Determination of Cellular Melanin Contents

To measure the melanin content, the B16F10 cells were seeded on a 60 mm cell culture dish and treated with various concentrations of samples over a 72 h period. The obtained cell pellet was treated with 200 µL of 1N NaOH containing 10% DMSO at 90 °C for 10 min. Then, it was transferred to a 96-well plate for absorbance measurements to be taken at 405 nm using a microplate reader (Biotek; Winooski, VT, USA).

### 4.5. Western Blotting

After seeding the B16F10 cells on a 60 mm cell culture dish and incubating them for 24 h, the cells were treated with 6-methylcoumarin (100, 200, and 250 µM) for 72 h. To obtain the protein, the cells were washed twice with cold 1× PBS and then lysed using a cell lysis solution containing an inhibitor cocktail. The resulting mixture was centrifuged at 15,000 rpm for 20 min to obtain a supernatant. The protein was then quantified and electrophoresed using SDS-polyacrylamide gel, transferred to a membrane, and blocked with 5% skim milk. After being washed six times with 1× PBS, the primary antibody was added and allowed to react overnight. The following day, the membrane was washed six times and the secondary antibody was added for a 2 h incubation period. The protein bands were then detected using an ECL solution and the band images were analyzed using ImageJ software (NIH, Bethesda, MD, USA).

### 4.6. Human Skin Irritation Test

A total of 33 healthy subjects without skin problems participated in the skin primary irritation test. The age of the subjects ranged from 25 to 53 years old and the average age was 45.82 ± 7.84. 6-methylcoumarin (125 and 250 μM) formulated with squalene was patched onto their skin for 48 h. After the patch was removed, the condition of their skin was evaluated according to the modified Frosch and Kligman and CTFA guidelines. The intensity of the skin’s reaction was measured twice at 30 min and 24 h after the patch was removed. This test was performed at the Dermapro Skin Science Research Center (Seoul, Korea) and was conducted according to the human test guideline regulations based on the Declaration of Helsinki (IRB no. 1-220777-A-N-01-DICN23044).

### 4.7. Statistical Analysis

The mean ± standard deviation (SD) of three independent experiments was used to present the experimental data. The statistical significance was determined using a Student’s *t*-test (IBM SPSS v. 20, SPSS Inc., Armonk, NY, USA) and was denoted as follows: # *p* < 0.001 vs. the unstimulated control group, * *p* < 0.05, ** *p* < 0.01, and *** *p* < 0.001 vs. the α-MSH treatment group alone.

## Figures and Tables

**Figure 1 molecules-28-04551-f001:**
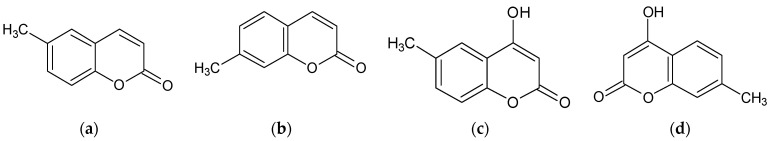
The chemical structures of 6-methylcoumarin (**a**), 7-methylcoumarin (**b**), 4-hydroxy-6-methylcoumarin (**c**), and 4-hydroxy-7-methylcoumarin (**d**). All chemical structures of the four compounds were re-drawn using ChemDraw Ultra 11.0 (Cambridge Soft Corporation, Cambridge, MA, USA).

**Figure 2 molecules-28-04551-f002:**
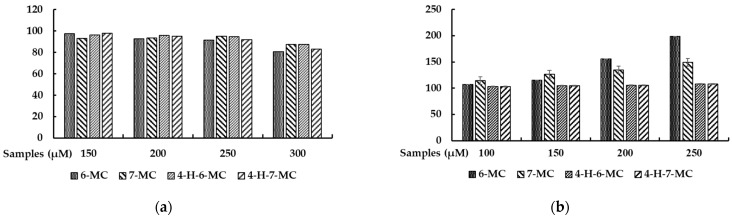
Effect of methylcoumarin derivatives on cell viability (**a**) and melanin production (**b**) in B16F10 cells. The cells were treated with methylcoumarin derivatives (100, 150, 200, 250, or 300 μM) for 72 h. The results are presented as the mean ± SD from three independent experiments.

**Figure 3 molecules-28-04551-f003:**
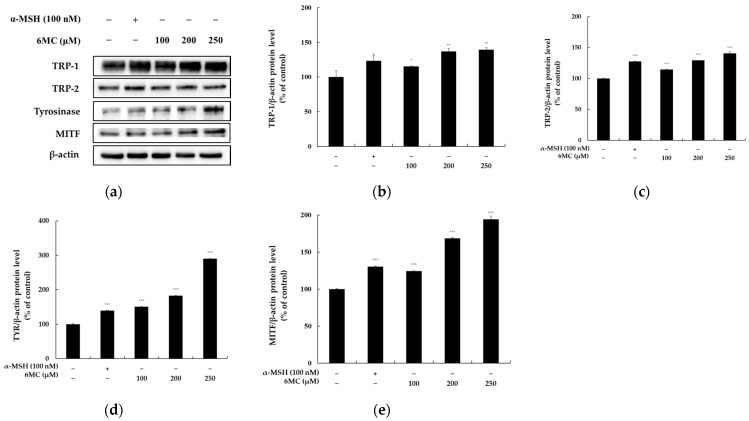
Effect of 6-methylcoumarin on the melanogenesis-related protein expression of B16F10 cells. Cells were treated with 6-methylcoumarin at indicated concentrations and were then harvested. (**a**) indicates protein expression by Western blot. The TRP-1 (**b**), TRP-2 (**c**), TYR (**d**), and MITF (**e**) protein levels were evaluated using Western blot. The relative intensity of the protein band was quantified using ImageJ software and the value was normalized to that of the corresponding loading control. Untreated cells were regarded as 100%. *** *p* < 0.001, ** *p* < 0.01, and * *p* < 0.05 as compared to the control group.

**Figure 4 molecules-28-04551-f004:**
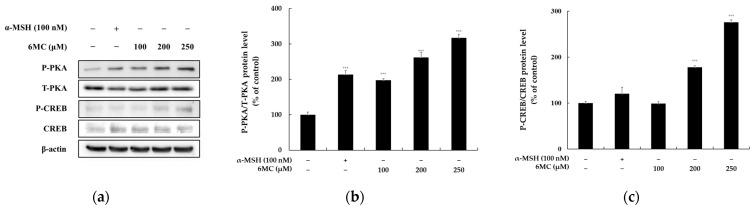
Effect of 6-methylcoumarin on PKA and CREB expression in B16F10 cells. Cells were treated with 6-methylcoumarin at indicated concentrations for 4h and then harvested. The protein and phosphorylation level of the pigmentation-related molecules was evaluated using Western blot. (**a**) indicates protein expression by Western blot. 6-methylcoumarin decreased P-PKA (**b**) and CREB (**c**) phosphorylation. Actin was used as a loading control. The relative intensity of the protein band was quantified using ImageJ software and the value was normalized to that of the corresponding loading control. The untreated cells were regarded as 100%. *** *p* < 0.001 as compared to the control group.

**Figure 5 molecules-28-04551-f005:**
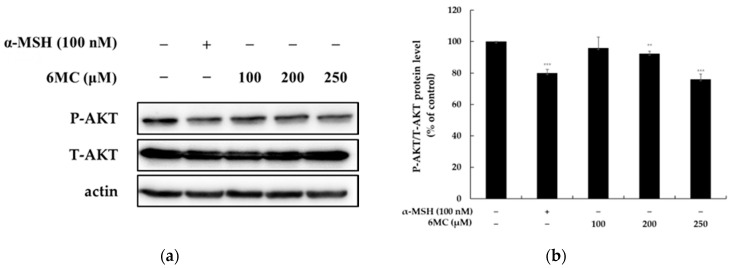
Effect of 6-methylcoumarin on the AKT protein expression of B16F10 cells. Cells were treated with 6-methylcoumarin at indicated concentrations for 4h and were then harvested. (**a**) Western blotting results and protein expression of (**b**) P-AKT/AKT. The relative intensity of the protein band was quantified using ImageJ software and the value was normalized to that of the corresponding loading control. Untreated cells were regarded as 100%. *** *p* < 0.001 and ** *p* < 0.01 as compared to the control group.

**Figure 6 molecules-28-04551-f006:**
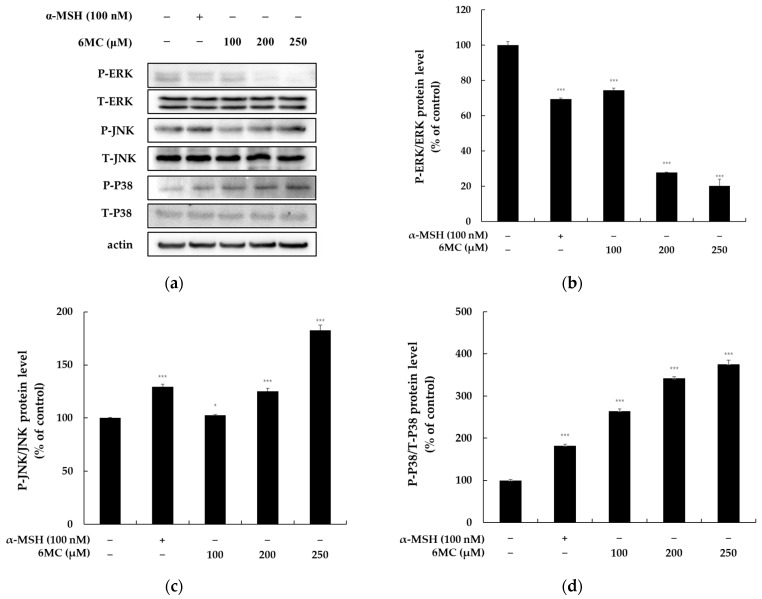
Effect of 6-methylcoumarin on ERK, JNK, and P38 expression in B16F10 cells. Cells were treated with 6-methylcoumarin at indicated concentrations for 4h and were then harvested. The protein and phosphorylation level of the pigmentation-related molecules was evaluated using Western blot. (**a**) indicates protein expression by Western blot. 6-methylcoumarin decreased the P-ERK phosphorylation (**b**). Reversibly, it decreased P-JNK (**c**) and P-P38 (**d**) phosphorylation. Actin was used as a loading control. The relative intensity of the protein band was quantified using ImageJ software and the value was normalized to that of the corresponding loading control. Untreated cells were regarded as 100%. *** *p* < 0.001 and * *p* < 0.05 as compared to the control group.

**Figure 7 molecules-28-04551-f007:**
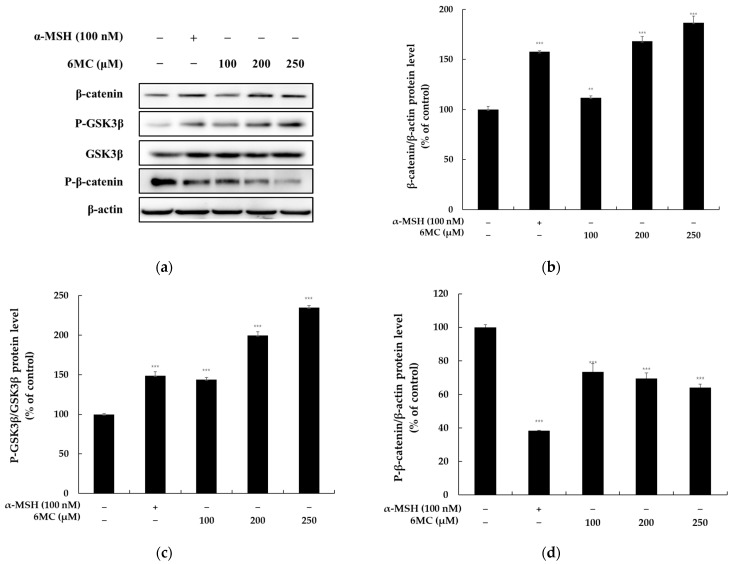
Effect of 6-methylcoumarin on β-catenin, P-GSK3β, and P-β-catenin expression in B16F10 cells. Cells were treated with 6-methylcoumarin at indicated concentrations for 24h and were then harvested. The protein level of the pigmentation-related molecules was evaluated using Western blot. (**a**) indicates protein expression by Western blot. 6-methylcoumarin increased the β-catenin protein levels (**b**) and P-GSK3β phosphorylation (**c**). Reversibly, it decreased P-β-catenin phosphorylation (**d**). Actin was used as a loading control. The relative intensity of the protein band was quantified using ImageJ software and the value was normalized to that of the corresponding loading control. Untreated cells were regarded as 100%. *** *p* < 0.001 and ** *p* < 0.01 as compared to the control group.

**Figure 8 molecules-28-04551-f008:**
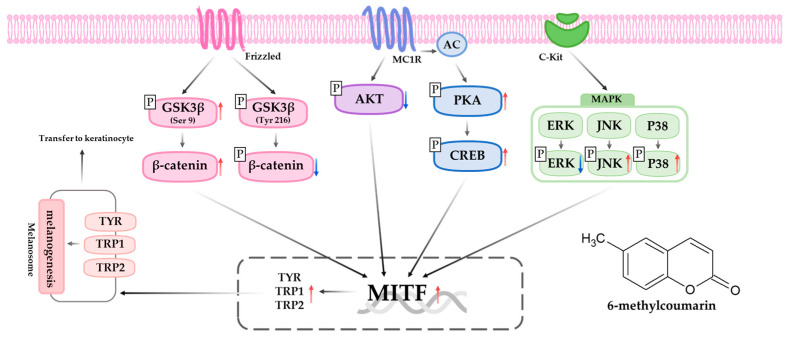
Schematic diagram of the proposed mechanism regulating the stimulative action of 6-methylcoumarin on melanogenesis. A blue arrow means decrease or inhibition, and a red arrow means increase or activation.

**Table 1 molecules-28-04551-t001:** The results from the primary human skin irritation tests (*n* = 33).

No.	Test Sample	No. of Respondents	20 min after Removal				24 h after Removal				Reaction Grade (R) ***		
			+1	+2	+3	+4	+1	+2	+3	+4	24 h	48 h	Mean
1	6-MC (125 μM)	0	-	-	-	-	-	-	-	-	0	0	0
2	6-MC (250 μM)	0	-	-	-	-	-	-	-	-	0	0	0
3	Squalene	0	-	-	-	-	-	-	-	-	0	0	0

* The irritation degree was graded as no-to-slight irritation, with scores ranging from 0.00 to less than 0.87.

## Data Availability

The data presented in this study are available on request from the corresponding author.

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
