# Peer review of "6-Methylcoumarin Promotes Melanogenesis through the PKA/CREB, MAPK, AKT/PI3K, and GSK3β/β-Catenin Signaling Pathways"

_molecules, 2023, doi:10.3390/molecules28114551_

Round 1

Reviewer 1 Report

1. Please mention the CAS Number and PubChem ID for each compound (four coumarin derivatives).

2. Figure 1, please state, is it self-produced? if yes, specify the software to produce the structure. If quoting from a publication, it must be clearly stated and cited.

3. In the section "4.7. Statistical Analysis" the type and software company used for the data analysis must be mentioned.

4. Very interesting, why does the "Data Availability Statement" = Not applicable ? even though it is clear that this manuscript is a research article which of course contains research data.

5. This study uses humans, but the "Informed Consent Statement" statement is Not applicable. this is very unclear and wrong.

6. This manuscript would be more interesting and improve readability if there were mechanistic/schematic figure that explaining 6-Methylcoumarin to promote melanogenesis via PKA/CREB, MAPK, AKT/PI3K, and GSK3β/β-Catenin signaling pathways.

Author Response

<Molecules-2423358>

< 6-Methylcoumarin Promotes Melanogenesis through the PKA/CREB, MAPK, AKT/PI3K, and GSK3β/β-Catenin Signal-ing Pathways>

Dear Editor,

Thank you for your useful comments and suggestions on the language and structure of our manuscript. We have modified the manuscript accordingly, and detailed corrections are listed below point by point (in red).

Reviewer #1

  1. Please mention the CAS Number and PubChem ID for each compound (four coumarin derivatives).:

→ We have provided the CAS numbers for the four compounds as requested.(L335-337)

  1. Figure 1, please state, is it self-produced? if yes, specify the software to produce the structure. If quoting from a publication, it must be clearly stated and cited.:

→ As the reviewer pointed out, we drew the structures of the four compounds ourselves and inserted the program information.(L98-100)

  1. In the section "4.7. Statistical Analysis" the type and software company used for the data analysis must be mentioned.:

→ As the reviewer pointed out, we mentioned the type of software and company used to analyze the data. (L394)

  1. Very interesting, why does the "Data Availability Statement" = Not applicable ? even though it is clear that this manuscript is a research article which of course contains research data.:

→ We didn't know what this meant, so we inserted something like this. “The data presented in this study are available on request from the corresponding author.” (L408-409)

  1. This study uses humans, but the "Informed Consent Statement" statement is Not applicable. this is very unclear and wrong.:

→ We wrote "IRB number: 1-220777-A-N-01-DICN23044".(L407)

  1. This manuscript would be more interesting and improve readability if there were mechanistic/schematic figure that explaining 6-Methylcoumarin to promote melanogenesis via PKA/CREB, MAPK, AKT/PI3K, and GSK3β/β-Catenin signaling pathways.:

→ We've included a mechanism illustration as recommended by the reviewer.(L329-332)

Reviewer 2 Report

In the current manuscript, Hyun et al. described the activity of coumarin derivatives in the process of melanogenesis. In scientific terms, the manuscript is an interesting study, and the research results are presented in a clear manner without substantive errors. My only criticism concerns the name of the molecule L-DOPA (page 2, line 49) - the correct name is L-3,4-dihydroxyphenylalanine (L-hydroxyl phenylalanine is a mental shortcut). With this minor correction, I support the publication of this manuscript in Molecules.

Author Response

<Molecules-2423358>

< 6-Methylcoumarin Promotes Melanogenesis through the PKA/CREB, MAPK, AKT/PI3K, and GSK3β/β-Catenin Signal-ing Pathways>

Dear Editor,

Thank you for your useful comments and suggestions on the language and structure of our manuscript. We have modified the manuscript accordingly, and detailed corrections are listed below point by point (in red).

Reviewer #2

1 In the current manuscript, Hyun et al. described the activity of coumarin derivatives in the process of melanogenesis. In scientific terms, the manuscript is an interesting study, and the research results are presented in a clear manner without substantive errors. My only criticism concerns the name of the molecule L-DOPA (page 2, line 49) - the correct name is L-3,4-dihydroxyphenylalanine (L-hydroxyl phenylalanine is a mental shortcut). With this minor correction, I support the publication of this manuscript in Molecules.:

→ We have corrected it to L-3,4-dihydroxyphenylalanine as pointed out by the reviewer. (L49)

Reviewer 3 Report

This article will be a nice addition to Molecules after the following issues have been addressed.

1)    It would be easier to interpret the biological data for the various compounds if the compounds were numbered (1-4 or a-d) and this numbering convention was used for the compounds in the text, figures and figure cations.

2)    The figures would be easier to read if one unified bar chart was added where the activity for all four compounds were directly next to each other.

Author Response

<Molecules-2423358>

< 6-Methylcoumarin Promotes Melanogenesis through the PKA/CREB, MAPK, AKT/PI3K, and GSK3β/β-Catenin Signal-ing Pathways>

Dear Editor,

Thank you for your useful comments and suggestions on the language and structure of our manuscript. We have modified the manuscript accordingly, and detailed corrections are listed below point by point (in red).

Reviewer #3

  1. It would be easier to interpret the biological data for the various compounds if the compounds were numbered (1-4 or a-d) and this numbering convention was used for the compounds in the text, figures and figure cations. The figures would be easier to read if one unified bar chart was added where the activity for all four compounds were directly next to each other:

→ We have revised the organization of the paper to incorporate Figure 1 and Figure 2 as pointed out by the reviewer.
